# Through-container quantitative analysis of hand sanitizers using spatially offset Raman spectroscopy

Nirzari Gupta[1], Jason D. Rodriguez[1] & Huzeyfe Yilmaz [1✉]

The COVID-19 pandemic created an increased demand for hygiene supplies such as hand sanitizers. In response, a large number of new domestic or imported hand sanitizer products entered the US market. Some of these products were later found to be out of specification. Here, to quickly assess the quality of the hand sanitizer products, a quantitative, through-container screening method was developed for rapid and non-destructive screening. Using spatially offset Raman spectroscopy (SORS) and support vector regression (SVR), active ingredients (e.g., type of alcohol) of 173 commercial and in-house products were identified and quantified regardless of the container material or opacity. Alcohol content in hand sanitizer formulations were predicted with high accuracy ($R^2 > 0.98$) using SVR and 94% of the substandard test samples were identified. In sum, a SORS-SVR method was developed and used for testing medical countermeasures used against COVID-19, demonstrating a potential for high-volume testing during public health threats.

[1] Division of Complex Drug Analysis, Office of Testing and Research, Office of Pharmaceutical Quality, Center for Drug Evaluation and Research, US Food and Drug Administration, St Louis, MO, USA. ✉email: huzeyfe.yilmaz@fda.hhs.gov

Hand hygiene plays an important role in the response to the current outbreak of corona virus disease 2019 (COVID-19) caused by severe acute respiratory syndrome coronavirus 2 (SARS-CoV-2)[1]. While washing hands with soap and water is recommended, when not available, Centers for Disease Control and Prevention (CDC) recommends using a hand sanitizer that contains at least 60% alcohol (ethyl alcohol or ethanol)[2]. To reduce the transmission of SARS-CoV-2 in public places where maintaining hand hygiene may be difficult, hand sanitizers have been used frequently, creating an increased demand for such products[3]. During the earlier stages of the COVID-19 pandemic, consumers and healthcare personnel faced challenges in accessing hand sanitizers[4].

In response, US Food and Drug Administration (FDA) provided a temporary guidance to firms that are not currently regulated as drug manufacturers, for preparation of alcohol-based hand-sanitizer products[5]. More recently, the variety of hand-sanitizer products available to the public have increased as many new products have become available from commercially available sources. However, certain products were found to contain less-than-required amounts of active-ingredient alcohols and/or toxic contaminants such as methanol and 1-propanol[6]. According to CDC, methanol-contaminated hand sanitizers (hand rubs) can be absorbed through skin and cause chronic toxicity[7]. CDC also reported incidents of methanol poisoning and even deaths associated with the ingestion of hand sanitizers since the pandemic has started[8]. While highly contaminated or adulterated products pose direct health risks, subpotent hand sanitizers can be inefficient in reducing the virus transmission from surfaces and create a false sense of safety for consumers. To maintain a high-quality hand-sanitizer supply, frequent testing is required. Development of a rapid screening method may allow swift regulatory action or prioritization of samples for further analysis. Potentially, highly contaminated, low-quality hand-sanitizer products[6] can be rapidly tested using through-container spectroscopic methods and removed from the supply chain expeditiously.

Previously, spectroscopic field applications have been developed for circumstances similar to the current global pandemic. Raman-based methods were developed for rapid screening of medical countermeasures such as anti-infectives[9] and stockpiled drugs[10] and other pharmaceuticals to identify counterfeits[11] and unapproved drug products[12]. Portable Raman methods were also developed for liquid-based samples where binary mixtures of organic solvents or the amount of ethanol in different alcoholic drinks were measured[13,14]. Further innovation in the field was demonstrated with introduction of spatially offset Raman spectroscopy (SORS), as a noninvasive through-container detection and identification tool[15–17].

Since diffuse light can reach subsurfaces under most circumstances, inelastically scattered Raman photons are in fact present. However, subsurface Raman photons cannot be easily detected at spatial incidence where surface Raman and fluorescence photons are present at higher numbers. To avoid this, a spatial offset can be used for collection of light and subsurface Raman features can be revealed using scaled subtraction of Raman spectra at incidence and spectra collected at a spatial offset[17–19]. Initially, SORS found applications in pharmaceutical analysis and explosive detection where the contents of a plastic or glass container were of interest, while preserving the sample integrity was crucial[15,16]. In a more recent example, Ellis et al. demonstrated alcohol content and impurity detection in drinking alcohols using a portable SORS instrument[20]. Although the availability of portable SORS devices paves the way for numerous applications[21], quantitative SORS has not been demonstrated in a universal application with container variety.

Note that throughout this paper, "traditional Raman" was termed for spectra collected at zero offset (incidence), "offset" for the spectra acquired at a spatial offset, and "scaled-subtracted" for their scaled subtraction. The term SORS was reserved for the technique to avoid ambiguity.

The basis for quantitative Raman lies within the relationship between the spontaneous Raman scattering intensity integrated over a particular band $I_R(v)$, the excitation intensity $I_L$, and the number of scattering molecules $N$[22,23]

$$I_R(v) \propto I_L\, N\, \frac{(v_0 - v)^4}{\left(1 - e^{-hv/kT}\right)} \tag{1}$$

where $v_0$ is the excitation frequency and $v$ is the vibrational frequency of the molecule. In through-container spectroscopy, the linearity between $I_R(v)$ and $N$ is disrupted by $I_L$ being reduced due to electromagnetic absorption, and elastic/inelastic scattering of light from container surface prior to interacting with the contents. Therefore, most quantitative models that rely on latent variables and covariances (linear models) would yield erroneous predictions and/or nonrobust models[24]. Although container-specific linear models can be created, they cannot provide practical and rapid analyses since industrial packaging materials vary in type and opacity. However, combination of subsurface probing via SORS and spectral postprocessing can provide adequate Raman data for prediction of the concentration of a particular analyte inside a container. Although SORS can penetrate most plastic and glass containers, its practical capabilities have not been fully explored with advanced chemometric tools. Since container material and manufacturing properties introduce various nonlinear effects for the above equation, a nonlinear chemometric modeling technique may be best suited for universal quantitative through-container analysis.

Herein, a SORS-based quantitative method was developed for through-container analysis of hand-sanitizer products. The superiority of SORS over traditional Raman was demonstrated using multivariate analysis (MVA) of spectra from hand-sanitizer formulations with four different alcohols. When various types of containers (plastic, glass, opaque, transparent, etc.) were used for each formulation, only ~21% of the variation in the traditional Raman dataset was representative of contents (~77% of the variation was representative of containers). On the other hand, more than 99% of variation in the scaled–subtracted dataset was representative of contents, resulting in clear separation of alcohols in the score space of components representing the alcohols in the formulations. Previously, quantitative SORS has only been shown under container uniformity[21]; here a method was developed to quantify the amount of alcohol in hand sanitizer solutions inside containers with varying opacity and material type. Quantification was achieved using a normalization step prior to regression and taking advantage of concentration-dependent effects in the Raman spectra of alcohol–water mixtures. By transforming the scaled–subtracted data to a higher-dimensional feature space using a Gaussian kernel, and performing regression with support vectors, quantification of alcohols was achieved with high accuracy (root mean-squared error (RMSE) ~ 2–5%). Quantitative results of SORS–SVR method were used in decision trees where contaminated, subpotent, or fair determinations were made with 90.7% accuracy. In general, false negatives and false positives were samples that contained alcohols below the limit of detection, which varied for each alcohol type and the formulation (1–5% v/v). Overall, a chemometric approach was utilized in SORS to enable quantification via through-container screening, regardless of the container thickness, material, or opacity. Overcoming the limitations (lack of specificity and consideration for matrix effects in finished products) of widely used library-search methods, SORS–SVR provided nondestructive and quantitative analysis of medical countermeasure products used against COVID-19.

## Results and discussion

**Through-container analysis of hand sanitizers using SORS.** A schematic representation of SORS is presented in Fig. 1a. Intensity distribution of collected photons was shown when a lateral offset was used (red and green arrows indicate the positions for incidence and collection positions, respectively). In traditional Raman, light is collected at the source position where the majority of the photons are those that travel a short distance. Hence, subsurface Raman features are usually overwhelmed by the surface fluorescence or Raman. For light collected at a lateral offset, photons that travel longer are likely to be detected as well. Therefore, inelastically scattered light collected at a spatial offset contains a higher portion of the subsurface Raman photons compared with traditional Raman[25,26]. By scaling and subtracting traditional and offset Raman spectra, one can obtain spectral features of subsurfaces[17]. Based on this principle, SORS has become a practical tool for through-container analysis of raw materials and screening of pharmaceuticals and other sensitive materials[15,16,20].

Traditional and offset spectra, and their scaled subtraction from a hand-sanitizer solution inside a semitransparent high-density polyethylene (HDPE) container, are shown in Fig. 1c. Since the container was translucent, the traditional spectrum had

features from both the polyethylene (PE) packaging material (1130 and 1296 cm$^{-1}$) and active ingredient alcohol–ethanol (882, 1049, and 1092 cm$^{-1}$) of the hand sanitizer at similar intensities (black). In the offset spectrum, ethanol features were more intense than the container features (red). After scaled subtraction and polynomial baseline correction, the resulting spectrum predominantly contained Raman features of ethanol. Although the semitransparent container used in Fig. 1c allowed observation of content features via traditional Raman, many packaging materials used for hand sanitizers were observed to be opaque (Fig. 1b). In this study, two separate alcohols, approved for hand-sanitizer formulations, ethanol and 2-propanol, and two commonly found contaminant/adulterant alcohols with different chemical structures (methanol and 1-propanol) were used[27]. Scaled–subtracted Raman spectra of hand-sanitizer formulations containing the four alcohols used in this study are shown in Fig. 1d. Based on literature comparisons, ethanol could be identified by 882 cm$^{-1}$ C–C stretching, 1049 cm$^{-1}$ C–O stretching, 1092 cm$^{-1}$ CH$_3$ rocking, and 1454 cm$^{-1}$ CH$_3$ bending modes[28]. Methanol could be identified by the 1030 cm$^{-1}$ C–O stretching mode[29]. 2-propanol was identified by the 818 cm$^{-1}$ C–C and 950 cm$^{-1}$ C–O stretching modes[30]. 1-propanol was identified by the 859 cm$^{-1}$ and 888 cm$^{-1}$ C–C stretching and 968 cm$^{-1}$ C–O stretching modes[31,32].

**Detection of contaminants and observation of spectral shifts in water–alcohol mixtures.** Spectra of binary mixtures of common contaminants and approved alcohols in hand-sanitizer formulations are shown in Fig. 2a. Methanol and 1-propanol could be identified in ethanol or 2-propanol-based hand-sanitizer formulations at concentrations as low as 2.5%. The effects of increasing concentrations of contaminant alcohol on the Raman spectra were shown and Raman bands representing each alcohol were marked with arrows (Fig. 2). At low concentrations, 1030 cm$^{-1}$ C–O stretching mode of methanol formed a shoulder in ethanol based hand sanitizers near the 1049 cm$^{-1}$ C–O stretching mode. 859 cm$^{-1}$ and 888 cm$^{-1}$ C–C stretching modes of 1-propanol caused broadening and formed a shoulder around the 882 cm$^{-1}$ C–C stretching mode of ethanol. On the other hand, the 968 cm$^{-1}$ C–O stretching mode of 1-propanol appeared without a background mode from the hand sanitizer formulations of ethanol or 2-propanol and the 1030 cm$^{-1}$ C–O stretching mode of methanol appeared without a background mode from the 2-propanol hand-sanitizer solution.

When the total alcohol amount in the hand-sanitizer solutions was varied, spectral peak shifts were observed. Spectra with varying alcohol amounts are compared in Fig. 2b, where changes in peak positions were indicated with arrows. Frequency shifts in C–O and C–C stretching modes of alcohols have been well documented previously[33–36] and were attributed to changes in the hydrogen bonding between alcohols and water. This effect was most clearly observed in methanol. A nonlinear relationship between the C–O stretching-mode frequency and concentration was previously reported and attributed to progressive hydration of hydroxyl and solvation of methyl groups[33]. It should be noted that, spectral shifts were not observed in alcohol mixtures with fixed total alcohol concentration (75 or 80% v/v), as the water content was not varied, supporting the hypothesis regarding hydrogen bonding and hydration. The amount of shifts in selected C–O and C–C bonds were analyzed as a function of alcohol amount in the hand-sanitizer formulation (Supplementary Fig. 1). This relationship was found to be nonlinear in nature and could be fitted with an exponential decay-type equation ($y = Ae^{-x/t}$).

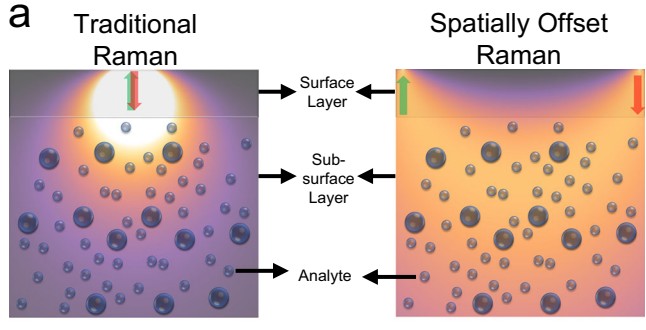

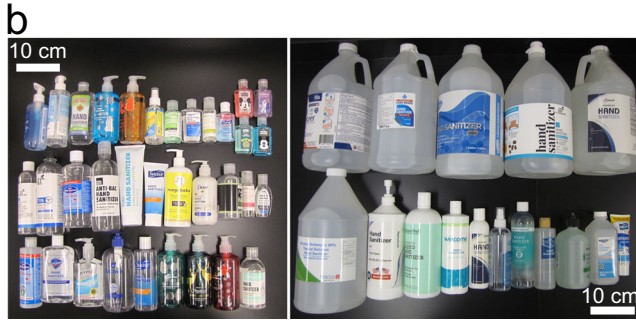

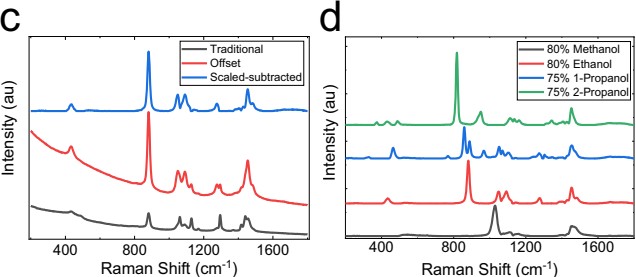

**Fig. 1 Through-container spectroscopy of hand sanitizers using SORS.**
**a** Schematic representation of the principle of spatially offset Raman. **b** Images from commercial hand sanitizer products used in the study. **c** Traditional and offset spectra, and their scaled subtraction from an ethanol-based hand-sanitizer solution. **d** Scaled–subtracted spectra of hand-sanitizers based on four alcohols used in the study.

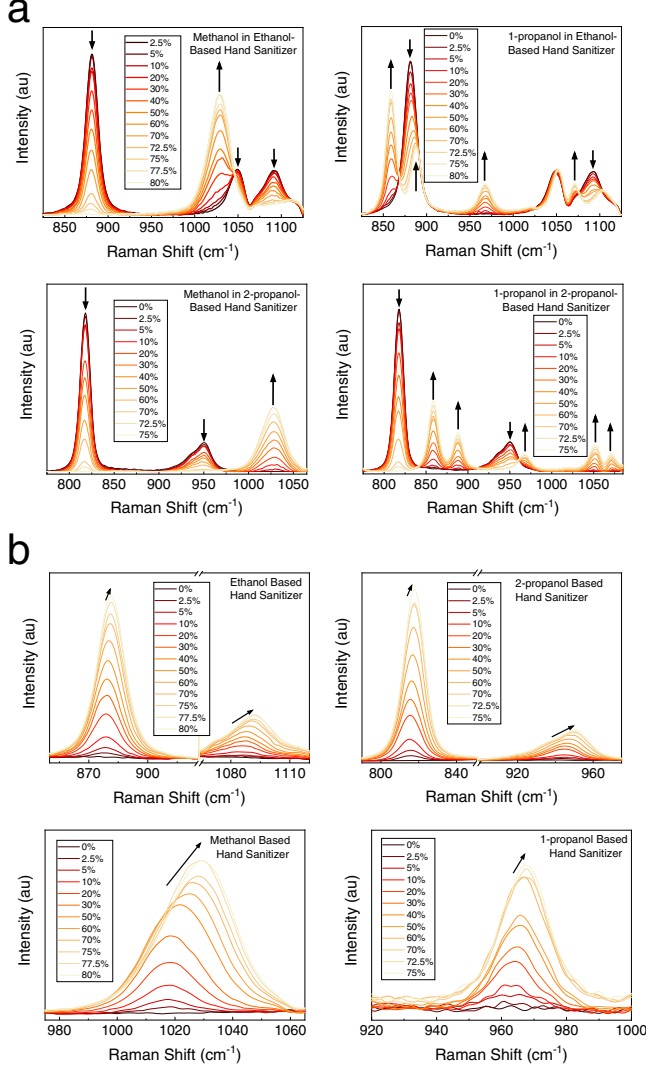

**Fig. 2 Raman spectra of contaminated and subpotent hand-sanitizer formulations.** Raman spectra of hand-sanitizer formulations with **a** ethanol, 2-propanol, methanol, and 1-propanol mixtures at a fixed total alcohol concentration and **b** varying amounts of alcohol and water.

**Statistical visualization of SORS–traditional Raman comparison**. Multivariate analysis is routinely employed for Raman datasets of chemical mixtures to infer otherwise obscure observations and perform quantitative analyses[37–42]. Among multivariate methods, multivariate curve resolution (MCR) has found excellent use in chemical mixture analysis due to its mathematical construction (see Supplementary Methods, Statistical analysis and modeling)[37,38]. To statistically demonstrate the effectiveness of SORS, offset and traditional Raman spectra of four alcohols (ethanol, 2-propanol, methanol, and 1-propanol) were collected in most commonly used packaging materials such as polyethylene (PE), polypropylene (PP), polyethylene terephthalate (PET), and glass at varying degrees of opacity. MCR was utilized for dimensionality reduction, visualization of variation in each dataset, and extraction of components that would resemble Raman spectra of containers and/or contents.

For traditional Raman spectra, a seven-component MCR model was designed (cumulative fit: 98.79%), since four alcohols and three types of containers with strong Raman signals were involved in the dataset (Fig. 3a–c). Each component either represented a container material or an alcohol. MCR components

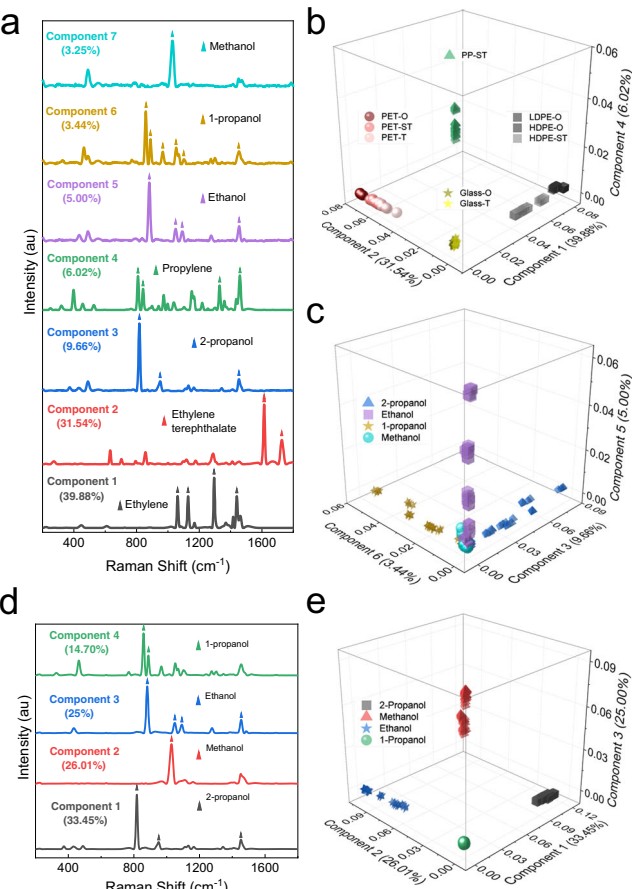

**Fig. 3 Statistical comparison of SORS and traditional Raman using multivariate curve resolution.** MCR scores and components of traditional (**a**–**c**) and scaled-subtracted (**d**, **e**) spectra of 2-propanol, 1-propanol, ethanol, or methanol-based hand sanitizers in various types of containers.

corresponding to container materials could be confirmed with specific Raman features of ethylene at 1064 cm$^{-1}$, 1130 cm$^{-1}$, 1296 cm$^{-1}$, and 1440 cm$^{-1}$, ethylene terephthalate at 1614 cm$^{-1}$ and 1728 cm$^{-1}$, and propylene at 1459 cm$^{-1}$[43–45]. Features in the MCR components were marked according to the material or alcohol Raman band (Fig. 3a). Components 1 (39.88%), 2 (31.54%), and 4 (6.02%) represented PE, PET, and PP with a total fit of 77.44%. When the scores were plotted on these three components, four clusters were observed, each corresponding to a certain type of container material. This is visualized in Fig. 3b, where the samples were color-coded based on container material and opacity (additional 2D score plots were provided in Supplementary Fig. 2). Furthermore, subclusters of PET and PE were observed, which correlated with the opacity of the container (O: opaque, ST: semitransparent, T: transparent). Traditional Raman data from more transparent containers were closer to the origin and vice versa, indicating a stronger influence of surface layers for materials with higher density, cross-linking, or thickness. Glass containers did not provide specific Raman bands associated with the weak glass signal and broad absorption/fluorescence bands were filtered out during preprocessing of the data. Hence, scores corresponding to glass containers were found to approach zero, near the origin in Fig. 3b. MCR components 3 (9.66%), 5 (5%), 6 (3.44%), and 7 (3.25%) represented the hand-sanitizer solutions based on 2-propanol, ethanol, 1-propanol, and methanol, respectively, with a total fit of 21.35%. When the scores were plotted on components 3, 5, and 6, they were spread along the three axes (Fig. 3c). However, for each alcohol, subclusters

due to container types were observed. Opposite to the opacity effect observed in Fig. 3b, data from more transparent containers were further from the origin (Supplementary Fig. 2b). Since MCR components 3, 5, 6, and 7 represented the contents, transparent containers allowed stronger content signal and therefore these data points were further from the origin compared with opaque containers.

For the scaled–subtracted spectra, a four–component MCR model was designed and a cumulative fit of 99.16% was computed. Features in the MCR components were marked with the corresponding Raman bands of alcohols where no features matching with container bands were observed (Fig. 3d). MCR scores of the scaled–subtracted data are visualized in Fig. 3e. Here, the samples were color-coded according to the type of alcohol used in the hand sanitizer formulation, similar to Fig. 3c. Unlike traditional Raman, scaled–subtracted spectra exclusively provided variation among alcohols. Overall, the effectiveness of SORS for through-container analysis was statistically demonstrated in the exercise summarized in Fig. 3. With traditional Raman, data could be fitted using components associated with container materials (77.44%) and contents (21.35%). When scaled–subtracted spectra were used, data could be fitted with components representing the contents (99.16%), despite having a wide variety of container materials and opacity. The separation among contents was clearly improved using scaled–subtracted spectra (Fig. 3e) compared with traditional Raman (Fig. 3c).

**Quantification of alcohols in hand sanitizers**. Last, development of a single quantitative model with container universality was sought. First, an adequate training set with a small sample size was designed by introducing variations due to container types/opacity and alcohol types/concentrations. The dataset used in the MCR analysis (Fig. 3) was utilized to account for container-induced effects while alcohol concentrations were kept constant. Data from binary mixtures of alcohols in hand sanitizer formulations with a fixed container type (semitransparent HDPE) were added. For binary mixtures, the total alcohol amounts were kept constant at the concentration recommended by the WHO[46]. The entire training set can be traced in Fig. 4. The test dataset consisted of 120 house samples and 53 commercial products, some of which are shown in Fig. 1c. The details of this dataset were presented in Supplementary Data 1.

Container material type (glass, plastic), thickness, additives used in the manufacturing (colorants, liners, etc.) and material-processing parameters (polymer molecular weight, cross-linking, etc.) determine the transmittance of light and affect the Raman features and their intensity in through-container spectroscopy. To minimize the intensity fluctuations and spectral distortions caused by variation in the containers, spectra were normalized during preprocessing (see Supplementary Methods, Algorithms and preprocessing). Although normalization alone cannot resolve the issue of quantification in through-container Raman spectroscopy, relative intensities of features may be utilized when more than one analyte is manifested in the spectrum.

Principal-component regression (PCR) and partial least squares (PLS) are very frequently used for quantitative analysis in vibrational spectroscopy[47–50]. Two separate PLS-regression models were constructed with four and eight latent variables (LV), based on evaluation of the cross-validation RMSE (Supplementary Fig. 3), Q-residuals and $T^2$ statistics (Supplementary Fig. 4). Predicted concentrations for each alcohol type were plotted against the measured or known concentrations based on the sample origin (training, test house and test commercial), container type (semitransparent, transparent, and opaque), and

number of alcohols present in the formulation (1 alcohol, 2 alcohols and 3 alcohols) in Fig. 4. Using the PLS model with four LVs (PLS-4), more than 97% of the total variation was captured; however, the model had unusual Q-residuals, indicating a large lack-of-fit for certain samples. The loadings for the first four LVs demonstrated the Raman features of all alcohols used in the study and provided separation among them, at 818 cm$^{-1}$ band of 2-propanol, 859 cm$^{-1}$ band of 1-propanol, 882 cm$^{-1}$ band of ethanol, and 1030 cm$^{-1}$ band of methanol (Supplementary Fig. 5). When the predictions using PLS-4 were color-coded based on the number of alcohols used in the formulation, it could be seen that the multi-alcohol formulations were predicted with less error than single-alcohol formulations (Fig. 4a–c). This was because the relative intensities of alcohols in mixtures contributed to the prediction in mixtures, whereas single-alcohol solutions were overpredicted since the spectra were normalized. The loadings for LVs five to eight indicated separation due to shifts in peak positions of the alcohols and a total of 2.11% of the variation in the training set was represented. Recall that spectral shifts were observed in water–alcohol mixtures (Fig. 2) and their dependence on the concentration was found nonlinear (Supplementary Fig. 1). The PLS model with eight latent variables (PLS-8) had reduced RMSE for cross-validation sets and predictions. In comparison with PLS-4, Q-residuals were reduced; however, large Hotelling's $T^2$ values were found, indicating that certain samples represented a large amount of variation as they were far away from the center of the model (Supplementary Fig. 4). Furthermore, using the PLS-8 model, certain 1-propanol-based single-alcohol formulations could not be predicted accurately (Fig. 4d–f). This could be attributed to the complex dependence of spectral shifts on the concentration in 1-propanol. In addition, when the predictions using PLS-8 were color-coded based on container opacity (Fig. 4e), it could be seen that the samples in transparent containers were overpredicted, whereas those in opaque containers were underpredicted. Given the container-related irregularities and formulation matrix-related spectral variations in hand sanitizers, PLS as a traditional linear-regression method was ineffective in through-container spectroscopy.

Due to their compatibility with nonlinear or small sample-size problems, support-vector machines (SVM) have found success in classification or regression applications in spectroscopy and microscopy[51–55]. Here, an SVM-based regression model was developed using the ε-insensitive algorithm with a Gaussian radial basis function (Grbf)[56–58]. The cost parameter, $C$ and gamma, $\gamma$ were optimized using a grid search (see Supplementary Note, Supplementary Fig. 6). The resulting predictions of all alcohols are shown in Fig. 4g–i. When compared with the PLS models, concentrations of alcohols in hand-sanitizer formulations were predicted with improved accuracy (Table 1). Reduced prediction and cross-validation RMSE and improved coefficient of determination ($R^2$) in SVR was attributed to the use of Grbf nonlinear kernel. Note that the spectral shifts of the C–O bands vs the alcohol amount in hand sanitizers also exhibited a nonlinear relationship (Supplementary Fig. 1). To confirm, spectra from well- (smaller prediction error) and poorly (larger prediction error) predicted hand-sanitizer products (highlighted in black and red squares, respectively in Fig. 4b, e, and h) in transparent, semitransparent, and opaque containers were selected. Spectra were compared before and after preprocessing (normalization and baseline correction with Whittaker filter, Supplementary Fig. 7). For well-predicted products, the spectral shifts due to hydrogen bonding in alcohol–water mixtures were clearly resolved. For products with larger prediction errors, the signal-to-noise ratio (SNR) was not large enough for the C–O band shift to be resolved clearly. It should be noted that reduced SNR in certain products could not be attributed to a single property of

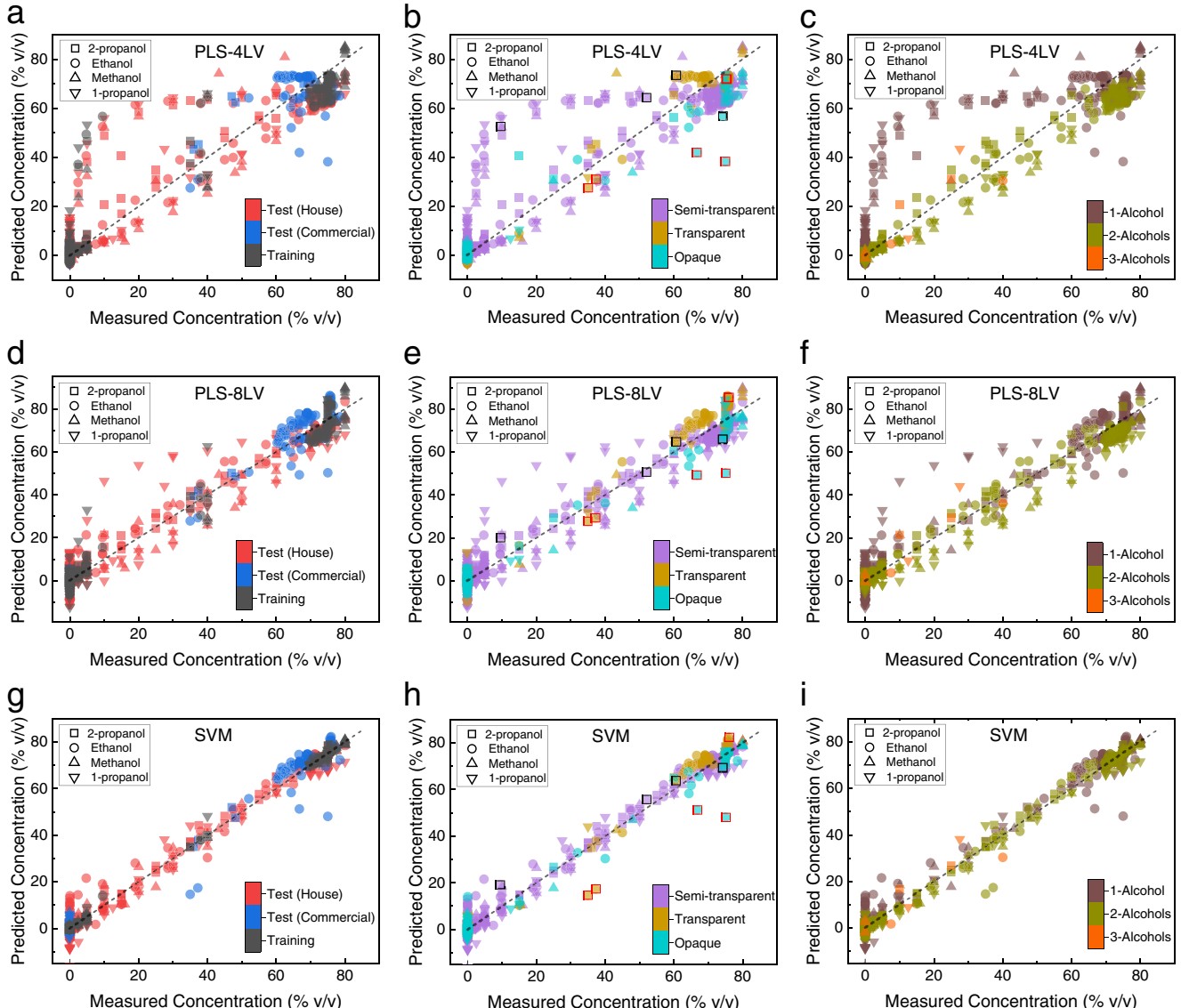

**Fig. 4 Quantification of alcohol amounts in hand sanitizers.** Predicted alcohol concentrations plotted against measured values for **a–c** PLS-4, **d–f** PLS-8, and **g–i** SVM models. Each symbol represents a separate alcohol, products are color-coded based on origin (**a**, **d**, **g**), container type (**b**, **e**, **h**), and the number of alcohols (**c**, **f**, **i**). Outliers with large prediction errors and well-predicted samples with small prediction errors are surrounded in red and black squares, respectively, and spectra are compared (Supplementary Fig. 7).

the containers as no direct relationship could be deduced based on thickness or material type.

There were also two products in transparent containers that were poorly predicted with SVR, which contained a total alcohol concentration of ~70%, consisting of about equal amounts of ethanol and 2-propanol. This error was expected, since the models were not trained for ethanol and 2-propanol mixtures, as it is unusual to use both of the recommended alcohols in a single hand-sanitizer formulation. The error for these products was less in PLS models, where loadings of several LVs contained separation for ethanol and 2-propanol. Note that a separate SVR model was computed for each alcohol used in the study, which is a limitation of the SVM method that can be addressed with training dataset design.

In total, 108 of the 173 test samples were substandard: 24 were subpotent and 84 were contaminated with 1-propanol and/or methanol. Predictions from SVM and PLS-regression models were used in a decision tree for semiquantitative determinations. Confusion matrices are shown in Fig. 5. The average accuracies of

the predictions were 77.5, 68.8, and 91%, for PLS-4, PLS-8, and SVM models, respectively. Samples with low amounts of contamination (less than 5%) and/or with potency at or around the minimum acceptable concentrations could be misclassified, given the limit of detection in through-container spectroscopy.

## Conclusions

In sum, coupling SORS with SVR yielded an efficient approach that could provide rapid and quantitative through-container analysis of alcohols in hand-sanitizer formulations. Here, quantitative analysis was performed with suitable accuracy for preliminary analysis while the samples remained unspoiled and unopened. This methodology can provide prioritization of adulterated or highly contaminated samples for testing with benchmark techniques. Currently, library search-based rapid screening methods are employed at various ports, borders, and security checkpoints. However, specificity, matrix effects, and quantification is beyond their reach. Quantitative SORS–SVR

method addresses the above issues in rapid screening, non-invasively and with container universality for a given product. A wide variety of hand-sanitizer products, such as gallon-size semitransparent containers, opaque plastic tubes, pouches and other types of containers were included in the analysis. Except for two highly opaque plastic containers (out of 53 commercial products), the concentration of alcohols in the formulations were predicted with RMSE in the range of 2–5% v/v, despite variations in the matrices of the products (liquid, gel, and additives used). The prediction RMSE of ethanol-based hand sanitizers was slightly less for liquid formulations (~3% v/v) compared with gel formulations (~4% v/v with outliers removed), however a fair comparison was not possible since the majority of commercial products were gel-based (Supplementary Fig. 8).

A quantitative spectroscopic method was developed for detection of impure or sub-standard hand sanitizer products using SORS. The advantage of using SORS over traditional Raman in through-barrier analysis was demonstrated with a multivariate statistical approach. Using a normalization step during preprocessing and utilizing non-linear regression methods (SVM), alcohol amount in the hand-sanitizer formulations could be accurately predicted, regardless of the type of container they were in. This was achieved by designing a training dataset with variation in containers as well as contents and taking advantage of concentration-dependent effects in alcohol–water mixtures. While quantification was achieved with product-specific container universality, samples remained unopened and unspoiled and the total time for analysis was only a few minutes. Hence, the SORS–SVR rapid screening technique developed here is suitable for public health emergencies, where an extremely large number of medical countermeasures such as drug products, sanitation supplies, or vaccines may need to be tested. As an alternative to library search-based qualitative rapid-detection methods used in field applications, SORS–SVR can provide specific and quantitative through-container analysis and has the potential to be widely adapted.

## Methods

**Reagents and materials**. HPLC-grade 2-propanol, LC–MS-grade methanol, ACS-grade glycerol, and hydrogen peroxide were purchased from Fisher scientific chemicals. USP-grade ethanol (95%) was purchased from Decon Laboratories, and ACS-grade 1-propanol was purchased from Sigma-Aldrich. LC–MS-grade purified water was used from Honeywell and ELGA PLFlex-2 DI water purifier with mΩ ≥ 18.0 at 25 °C. About 20-ml high-density polyethylene (HDPE), transparent, and amber glass scintillation vials were purchased from Fisher Scientific. Poly-ethylene terephthalate (PET), polyethylene (PE), and polypropylene (PP) containers of various colors were purchased from US plastic corporation.

**Instrumentation**. The Cobalt Light Systems (Agilent Technologies, Inc.) RapID, spatially offset (15 mm) Raman spectrometer was used throughout this study. The spectral range was 200–1800 cm$^{-1}$, power was 500 mW, and the wavelength was 830 nm. Laser power was utilized at 100%, acquisitions were 0.2 and 0.5 s, and the number of accumulations were 5 and 10 for zero (traditional) and offset spectra, respectively.

**Hand-sanitizer samples**. Commercial hand-sanitizer samples were purchased from national retail stores. Hand-sanitizer products available at the time were purchased and a total of 53 commercial products with varying formulations and packages were tested. In-house hand-sanitizer formulations were prepared according to the WHO formula recommendations and the FDA temporary policy[5,46]. Briefly, solutions with final concentrations of either 80% v/v ethanol or 75% v/v 2-propanol in aqueous solutions with 1.45% v/v glycerin and 0.145% v/v hydrogen peroxide were prepared. To simulate substandard (contaminated/substituted or subpotent) hand sanitizers, in-house samples were prepared by adding methanol and/or 1-propanol. Nine different containers were used for method development, testing, and storing of in-house hand-sanitizer formulations (Supplementary Fig. 9).

**Data acquisition**. Offset and traditional spectra were baselined (polynomial) and scaled–subtracted using the built-in options in RapID. For MCR analysis, both traditional and scaled–subtracted spectra were used. For partial least squares (PLS) and SVM-regression models, only scaled–subtracted spectra were utilized.

---

**Table 1 RMSE and coefficient of determination ($R^2$) of cross-validation and prediction for PLS models with four and eight latent variables, and SVR.**

| Alcohol | Cross-validation | | Prediction | |
|---|---|---|---|---|
| | RMSE (% v/v) | $R^2$ | RMSE (% v/v) | $R^2$ |
| | PLS-4 LV | PLS-4 LV | PLS-4 LV | PLS-4 LV |
| | PLS-8 LV | PLS-8 LV | PLS-8 LV | PLS-8 LV |
| | SVM | SVM | SVM | SVM |
| Ethanol | 10.09 | 0.882 | 9.56 | 0.912 |
| | 4.23 | 0.989 | 5.57 | 0.973 |
| | 2.09 | 0.995 | 4.23 | 0.986 |
| Methanol | 6.38 | 0.857 | 7.86 | 0.874 |
| | 3.92 | 0.979 | 4.06 | 0.972 |
| | 2.15 | 0.995 | 2.04 | 0.992 |
| 2-propanol | 7.62 | 0.932 | 7.51 | 0.914 |
| | 2.99 | 0.984 | 2.30 | 0.991 |
| | 2.05 | 0.995 | 1.52 | 0.997 |
| 1-propanol | 9.44 | 0.897 | 8.78 | 0.834 |
| | 7.77 | 0.93 | 7.73 | 0.867 |
| | 3.05 | 0.995 | 2.95 | 0.982 |

---

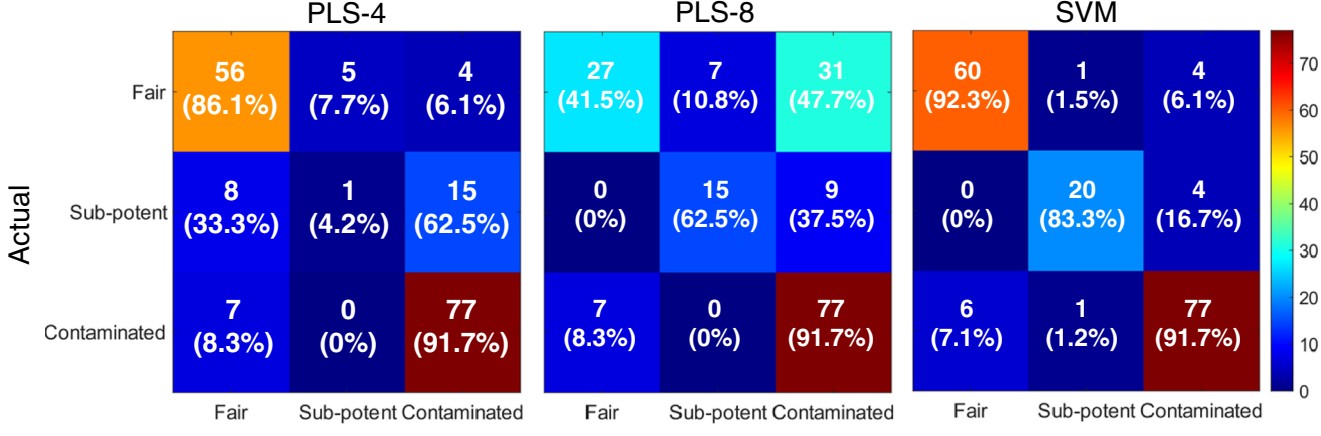

**Fig. 5 Classification of hand sanitizers based on quantification by SORS–SVR.** Confusion matrices for "fair", "sub-potent", and "contaminated" determinations based on PLS-4, PLS-8, and SVM models with overall accuracies of 0.77, 0.69, and 0.91.

**Preprocessing**. All spectra were baseline-corrected using automatic Whittaker filter and normalized by area using MATLAB R2020a (The MathWorks Inc., Natick, MA) and PLS Toolbox 7.9 (eigenvector Research Inc., Manson, WA).

**Statistical analysis**. Spectral preprocessing and multivariate analysis were performed in MATLAB R2020a (The MathWorks Inc., Natick, MA) and PLS Toolbox 7.9 (eigenvector Research Inc., Manson, WA). For the MCR models, five measurements were made for each combination of alcohol and container type. Hand-sanitizer formulations with primary alcohols at concentrations of 75% (for 2-propanol and 1-propanol) and 80% (for ethanol and methanol) were included. With four alcohols and nine containers, a total of 36 content–container combinations were formed. For SVR and PLS models, binary mixtures of approved and contaminant alcohols as well as subpotent hand-sanitizer solutions were introduced into the training library in addition to the content–container combinations described above (see Supplementary Data 1). An average of five measurements were used and a total of 78 spectra were included in the training dataset. Predictions were assessed based on the alcohol amount % v/v measured using a transmission Raman spectroscopy method (Supplementary Fig. 10) for commercial products and known concentrations of in-house test samples. PLS models were built using Q-residual and $T^2$ statistics for robustness and cross-validated using venetian blinds with 39 data splits of thickness one. SVR model was built using the ε-insensitive formulation with Gaussian radial basis function. Parameters for cost ($C$) and gamma ($\gamma$) were optimized using a grid search.

## Data availability
The data that support the findings of this study are available from the corresponding author upon reasonable request.

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

## Acknowledgements

This article reflects the views of the authors and should not be construed to represent FDA's views or policies.

## Author contributions

J.D.R. and H.Y. conceived the idea, H.Y. and N.G. designed the experiments, H.Y. and N.G. performed the experiments, and H.Y. and N.G. performed statistical analysis and data interpretation. J.D.R., H.Y., and N.G. discussed the results and wrote the paper.

## Competing interests

The authors declare no competing interests.
