## [Peer Review File · Communications Chemistry]

Reviewers' comments:

Reviewer #1 (Remarks to the Author):

The article "Through-Container Quantitative Analysis of Hand Sanitizers Using Spatially Offset Raman Spectroscopy" by Gupta et al presents a new application of SORS for the detection and identification of alcohol content in hand sanitizers. The manuscript is highly technical. The concept of SORS is already published in the literature. The novelty of the presented chemometrics models is moderate. The work may not be of wide interest to the readership of the journal and therefore may be more suitable for other technical/methods journals

Reviewer #2 (Remarks to the Author):

This paper reports on the use of spatially-offset Raman spectroscopy (SORS) for the through-container quantitative analysis of hand sanitizers. The paper is well written and the data is sound, and demonstrates that SORS has the ability to quantitatively analyze hand sanitizers even through opaque containers. I believe this work will be of interest to the Raman spectroscopy community in general, and represents an interesting advance in the area of quantitative SORS analysis.

While eventually publishable, there are several areas where the publication could be improved, and these are noted below:

Major concerns:

1. The authors indicate several spectral processing algorithms are used - the details of these algorithms should be provided in the SI.
2. The authors comment that peak shifts due to the contaminants are observed, this point requires further discussion and clarification.
3. The authors need to better clarify what is meant by zero offset / SORS / offset - the figure labels are not entirely consistent with the language used in the manuscript. Zero spectra and zero offset refer to "traditional" Raman?
4. Most of the Figures provided in the manuscript are of poor resolution and contain errors. Figure 1 is particularly poorly done. For example, it is not clear what Figure 1A is meant to represent, Figure 1C is hard to make out, Figure 1B and D provide pixelated spectra. In general, all of the figures require attention.
5. The authors used several commercial hand sanitizers - how was the quantitative analysis of these products completed, other than the SORS analysis? Certainly the authors didn't simply rely on the manufacturers label? I assume that HPLC / GC work was completed for these, but this information is missing from the manuscript. The authors should provide some clarification on this point.
6. Many of the commercial hand sanitizers were a gel - can the authors comment on the accuracy of their method for a liquid versus a gel?

Minor concerns:

1. In the experimental section, the resistivity of the water should be listed as $>18 \text{ MOhm}\cdot\text{cm}$
2. Several spelling errors appear throughout. A careful spell check is recommended.

Reviewer #3 (Remarks to the Author):

The author Gupta, Rodriguez and Yilmazt report interesting use of Spatially Offset Raman Spectroscopy to obtain a quick assessment thought container of hand sanitiser product (US Market). Overall the work is well written. The presented models for the prediction of alcohol percentage, based on SORS data of an extended number of home-based samples, are built on different multivariate analysis. However, I am not sure they have done themselves justice without overselling their claims. The authors claim the development of a container-universal method for quantitative analysis. On one hand, this methodology (SORS+ MCR, PCS, PLS etc..) was never applied for the assessment of hand sanitiser (the up to date methodology approved by FDA consist of GS-MS analysis) but hand Raman system and FT-Raman were already used for screening ethanol and methanol content in products and the proposed approach is well-know and already applied in other fields such as biomedical, pharmaceutical, cultural heritage. In this sense, I feel that their findings are novel only for a very specific case (hand sanitiser – quality check) but I am not sure the outcome of the work will be of interest to the extended reader of Communication Chemistry. I think the work is suitable for publication in an Applied Chemistry journal.

Moreover, the following point needs to be addressed before submission:

1. The author should review the discussion of their claim in the context of the previous literature, in particular:

a. In the last section of the introduction, the author summarizes their good results. On the contrary, the author should report the novelty of their approach (specific acquisition - data analysis performed) in the light of the previous literature

b. Revise text from line 87 to 96. It is written to be more likely a part of the abstract or conclusion. The final part of the introduction should stress the proposed approach, the comparison performed with the golden standard routine and the overall advantage, not the results obtained! That will help the reader to understand the logical steps of the authors while is reading the results

i. Proof of SORS advantage analyzing a mixture of contaminant with fixed %

ii. MA on the mixture on a different type of container (PE, PP glass)

iii. Comparison with a different parameter of the PLS, SVM models

iv. How is it achieve this insensitive to the container?? Is it the use of scale subtracted spectra? Need to be clearly stated

c. At the end of the introduction when the author presents their approach, they should refer to the limit of detection do Raman for the contaminant (concentration < 5 %) in comparison with the GS-MS. Is It enough for the analyzed system?

d. Line 77 to 82, the author should add a reference supporting the claim of the problematic problem for the semi-quantitative model (due to self-absorption from both container or material itself)

e. Previous literature should be ref more properly, need to add references in the following statements:

i. Line 43 - Ref. previous work with the golden standard for purity testing of hand sanitisers (according to FDA, GS-MS) – Add a brief comparison with results that may be obtained with golden standard to highlight the advantage of the proposed approach.

ii. Line 60-63 add a ref describing SORS principle and scale subtraction protocol. Refers properly the technique and the process first described by Matousek et al, therefore add the following:

a. Mosca, S.; Conti, C.; Stone, N.; Matousek, P. Spatially offset Raman spectroscopy, *Nat. Rev. Methods Prim.* 2021, 1 (1), 21.

b. Matousek, P. et al. Subsurface probing in diffusely scattering media using spatially offset Raman spectroscopy. *Appl. Spectrosc.* 59, 393–400 (2005).

2. The author should clarify within the text (results, method, SI..) what they mean with SORS data. Traditionally with the term, SORS means the Raman spectra acquired with spatial offset with respect to the excitation point. By reading the text I am not sure if the author instead with SORS means the scale subtraction between SORS and zero offsets. This must be clarified and eventually corrected.
3. In the same way, it is not clear if the approach was applied separately to the zero offset and spatial offset spectra or the scaled subtracted spectra. Can the author retrieve the native spectra or just the scaled one?
4. Add if possible the information on the spatial offset (fixed, I suppose) used by the instrument. In the instrumentation section, it seems that a bunch of pre-processing were automatically applied to the acquired spectra (scaled subtraction and polynomial baseline subtraction), but then in the statistical analysis section seems that further pre-processing (baseline correction) is applied before Multivariate analysis. Please Clarify if MCR, PLS, SVM were applied on the scaled subtracted spectra or the raw acquired one. If so, take into account that the container contribution is always subtracted and how a fluctuation of thickness may affect the prediction of alcohol percentage.
5. In S1 - together with the container picture, add geometrical information of the cylinder (radius, height, thickness), useful information for understanding the eventual effect on photo migrations. In particular, add a rough range of the container thickness with a comment on how is the proposed model affected by a variation of container thickness. Since the author aimed for a universal method, this is an important aspect.
6. Revise paragraph from line 152 to 155: the author used the wrong terminology for explaining the concept of SORS and sublayer sensitivity due to photon migration. The correct terminology should be used clearly. Specifically, it is not the incident light that is spread differently between the conventional Raman and SORS: if the excitation geometry is the same, the electromagnetic field and the intensity distribution of incident light in the media is the same. What is different is the capability to collect photons that propagate deeper (but not the incident light!!!). With the zero spatial offsets, the majority of the collected photons are statistically the ones that propagate for shorter distance (therefore surface information). With the use of spatial offset, it is possible to detect a larger number of photons that travel statistically deeper in the media (subsurface + surface). I suggest the author be more clear and if they required a deep knowledge on the subject I suggest reading the following reference on general photo migration (Martelli, F.; Binzoni, T.; Pifferi, A.; Spinelli, L.; Farina, A.; Torricelli, A. *Sci. Rep.* 2016, 6 (1), 27057.) or specifically SORS (Mosca, S.; Dey, P.; Tabish, T. A.; Palombo, F.; Stone, N.; Matousek, P. *Anal. Chem.* 2019, 91 (14), 8994–9000.)
7. Specify in Figure 1.a, what the 2D- intensity distribution map refers to – is it a simulation of the collected photons..?
8. The goal of the paragraph that starts from line 189 is not clear until the end of the paragraph (line 204 to 206). As it is, it is difficult to follow the presented results, I suggest to:
 - a. Moving up (at the start of the paragraph) “spectra of binary mixtures of common contaminants and approved alcohols in hand sanitiser formulations in semi-transparent containers was shown”.
 - b. to highlight in figure 2 the characteristic band of methanol, propanol and ethanol that were mentioned for the identification.
9. In SI, while introducing MCR, please mention the non-negativity constrain for the loadings – that’s why they are easily attributed to the components.
10. Figure 3, “MCR results”. The results not readable:
 - a. panel from A-C) convert the 3D score plot into two 2D panels (eg. PC1 VS PC2, PC1 VS PC4 or a similar combination pc2 vs pc4 etc) will improve the vision of eventual clustering especially on the origin – place the legend on a side
 - b. panel from D to E is related to the same MCR on the traditional Raman (it was not clear from the text and the sub-division) all the 7 components should be presented in the order of the same panel.

The author should try to use a colour scale for the spectra that do not confuse the reader between the scores of the MCR and the actual measurements (dots in the scores plot), which are actually a combination of the different scores with the proper coefficients.

c. Add the marks of the mentioned containers' Raman peak in the loadings (ref to line260-261).

d. In a similar way, panel C/F (related to MCR applied on SORS spectra) should contain all different loadings (7?) or just the 3 used for the score plot. I understand why the display also component 4, but the author should make a choice and keep the consistency. Eventually, the whole loadings and 2D scores plot can be added in SI.

11. The authors do not describe how the approach is quantitative. They clearly perform a calibration model with a specific % of Alcol but then, it is not clear what are markers used for the quantification. Do they use nominalized intensity?

12. At the end of the results/conclusion section, it was not discussed how the proposed approach manage to deal with a different container, the author simply stated the different accuracy in presence of more transparent or opaque media. Reading the abstract, I have expected a universal solution to the problem or a physical rationalize on the basis of the performed analysis to remove the container contributions. Instead, this is missing. The author should be careful about the claim stated in the abstract/introduction and conclusion. Moreover, the authors do not comment on relative intensity distortion due to absorption that can cause a fluctuation in the model, they do not refer to multiple problems that can't allow a direct quantitative analysis.

Response to Reviewers

Reviewer 1

The article “Through-Container Quantitative Analysis of Hand Sanitizers Using Spatially Offset Raman Spectroscopy” by Gupta et al presents a new application of SORS for the detection and identification of alcohol content in hand sanitizers. The manuscript is highly technical. The concept of SORS is already published in the literature. The novelty of the presented chemometrics models is moderate.

We thank the reviewer for their feedback. Even though our work has highly technical aspects regarding application of chemometrics, we believe that our method can be generalized to analysis of other pharmaceutical products and medical countermeasures. Rapid screening is commonly used at ports, borders, and security checkpoints with library search-based algorithms for identification of materials in packages and containers. Although spectral correlation-based identification is effective in analyzing unknown materials, it lacks specificity, ignores matrix effects for finished products and can become ineffective. The main novelty of our study is coupling spatially offset Raman spectroscopy (SORS) with advanced chemometric tools to overcome an important bottleneck in rapid screening: Through- container quantitative analysis with container-universality.

Reviewer 2

This paper reports on the use of spatially-offset Raman spectroscopy (SORS) for the through-container quantitative analysis of hand sanitizers. The paper is well written and the data is sound, and demonstrates that SORS has the ability to quantitatively analyze hand sanitizers even through opaque containers. I believe this work will be of interest to the Raman spectroscopy community in general, and represents an interesting advance in the area of quantitative SORS analysis.

While eventually publishable, there are several areas where the publication could be improved, and these are noted below:

We thank the reviewer for their comments. We have now revised our manuscript and improved the areas highlighted by the reviewer.

Major concerns:

1. The authors indicate several spectral processing algorithms are used - the details of these algorithms should be provided in the SI.

We thank the reviewer for this observation. We added a separate paragraph in the methods section with a reference to the SI where more detailed information was provided. Briefly, there are two steps of

preprocessing. During data acquisition, a built-in baseline subtraction and scaled subtraction of offset and traditional spectra was performed. Secondly, prior to application of regression methods, a Whittaker filter was applied, and spectra were normalized.

2. The authors comment that peak shifts due to the contaminants are observed, this point requires further discussion and clarification.

We thank the reviewer for this comment. We have expanded our discussion, provided additional figures and analysis (Figure 2A, 2B and S2) on this issue.

3. The authors need to better clarify what is meant by zero offset / SORS / offset - the figure labels are not entirely consistent with the language used in the manuscript. Zero spectra and zero offset refer to "traditional" Raman?

We thank the reviewer for recognizing this ambiguity. We have now clarified the terminology regarding zero, offset and SORS. We have corrected the figure labels to make the language consistent throughout the manuscript.

4. Most of the Figures provided in the manuscript are of poor resolution and contain errors. Figure 1 is particularly poorly done. For example, it is not clear what Figure 1A is meant to represent, Figure 1C is hard to make out, Figure 1B and D provide pixelated spectra. In general, all of the figures require attention.

We thank the reviewer for bringing the issue of figure quality to our attention. We have now revised our manuscript with higher quality figures. We have also revised Figure 1A to represent the effect of spatial offset more clearly in collection.

5. The authors used several commercial hand sanitizers - how was the quantitative analysis of these products completed, other than the SORS analysis? Certainly the authors didn't simply rely on the manufacturers label? I assume that HPLC / GC work was completed for these, but this information is missing from the manuscript. The authors should provide some clarification on this point.

In our revised manuscript, we have included results from 173 samples, of which 53 were commercial hand sanitizers. Due to various matrix effects (liquid vs gel, pH, etc.), development and validation of GC-MS methods for liquid and gel formulations are in progress and the results are yet to be disseminated. We have instead developed a quantitative transmission Raman spectroscopy method with container uniformity to determine the alcohol content in hand sanitizers. Note that compendial quantitative Raman methods (non-SORS) can be used for pharmaceuticals with qualified instruments and validated methods.¹

In our revised manuscript, we have utilized a highly accurate transmission Raman method to quantify the concentration of alcohols in commercial samples (Figure S3) and determined that most samples had alcohol amounts within the expected range (NMT 10% above, NLT 10% below the label claim) (Figure S4).

6. Many of the commercial hand sanitizers were a gel - can the authors comment on the accuracy of their method for a liquid versus a gel?

We thank the reviewer for allowing us to expand the discussion on the matrix and possible effects. We have now included the below figure in the supporting information, which demonstrates the effect of formulation on the prediction accuracy (liquid vs gel). We have also calculated our models and predictions separately for liquid gel formulations. We found that the RMSE of predictions for gel formulations were ~4 % v/v whereas for liquid formulations it was ~3 % v/v. We have now revised our manuscript and added a discussion on this issue.

Minor concerns:

*1. In the experimental section, the resistivity of the water should be listed as >18 MOhm*cm*

We fixed this error in our revised manuscript.

2. Several spelling errors appear throughout. A careful spell check is recommended.

We fixed the spelling errors in our revised manuscript.

Reviewer 3

The author Gupta, Rodriguez and Yilmazt report interesting use of Spatially Offset Raman Spectroscopy to obtain a quick assessment thought container of hand sanitiser product (US Market). Overall the work is well written. The presented models for the prediction of alcohol percentage, based on SORS data of an extended number of home-based samples, are built on different multivariate analysis. The presented models for the prediction of alcohol percentage, based on SORS data of an extended number of home-based samples, are built on different multivariate analysis. However, I am not sure they have done themselves justice without overselling their claims. The authors claim the development of a container-universal method for quantitative analysis. On one hand, this methodology (SORS+ MCR, PCS, PLS etc..) was never applied for the assessment of hand sanitiser (the up to date methodology approved by FDA consist of GS-MS analysis) but handheld Raman system and FT-Raman were already used for screening ethanol and methanol content in products and the proposed approach is well-know and already applied in other fields such as biomedical, pharmaceutical, cultural heritage. In this sense, I feel that their findings are novel only for a very specific case (hand sanitiser – quality check) but I am not sure the outcome of the work will be of interest to the extended reader of Communication Chemistry. I think the work is suitable for publication in an Applied Chemistry journal.

We thank the reviewer for acknowledging that coupling SORS with SVM has not been demonstrated before and indicating that handheld Raman and FTIR instruments are routinely employed for rapid screening in the field. However, we would like to point out that the current practices with handheld devices are based on library search methods which lack specificity, ignore matrix effects for finished products and can become ineffective due to variation in container materials and opacity. Our methodology provides a route to overcome the above issues and it allows fast-tracked quantification (both for certain adulterants and active ingredients) that can be used to identify and prioritize products for further testing or regulatory action.

In our revised manuscript, we have included results from 173 samples, of which 53 were commercial hand sanitizers. A wide range of formulations (gel, liquid, scented, etc.) in various types of containers were tested (PET, PE, translucent, opaque, pouch, gallon-sized containers, etc.). The reason for including a large number of in-house products was to ensure the SORS-SVR method was tested for quantification of contaminants and identification of sub-potent samples.

Although our proof of principle method is applied to hand sanitizers, our methodology (designing a training set that represents variation in container materials as well as variation in contents; using advanced chemometrics and machine learning tools for quantification) can be generalized to other types of medical countermeasures and pharmaceutical products. Our method allows through-container, rapid and quantitative testing without opening or spoiling products. During a public health threat where millions of doses of drugs and vaccines need to be manufactured and distributed, the importance of our method can be more clearly understood with the need for high volume testing while preserving product integrity.

1. The author should review the discussion of their claim in the context of the previous literature, in particular:

We thank the reviewer for allowing us to improve the discussion in the introduction.

a. In the last section of the introduction, the author summarizes their good results. On the contrary, the author should report the novelty of their approach (specific acquisition - data analysis performed) in the light of the previous literature

The last section of the introduction in our revised manuscript now includes the novelty of our approach with the specific Raman variant SORS and our data analysis: *“Herein, a SORS-based quantitative method was developed for through-container analysis of hand sanitizer products. The superiority of SORS over traditional Raman was demonstrated using multivariate analysis (MVA) of spectra from hand sanitizer formulations with four different alcohols. When various types of containers (plastic, glass, opaque, transparent, etc.) were used for each formulation, only ~21% of the variation in the traditional Raman dataset was representative of contents (~77% of the variation was representative of containers). On the other hand, more than 99% of variation in the scaled-subtracted dataset was representative of contents, resulting in clear separation of alcohols in the score space of components representing the alcohols in the formulations. Previously, quantitative SORS has only been shown under container uniformity²¹, here a method was developed to quantify the amount of alcohol in hand sanitizer solutions inside containers with varying opacity and material type. Quantification was achieved using a normalization step prior to regression and taking advantage of concentration dependent effects in the Raman spectra of alcohol-water mixtures. By transforming the scaled-subtracted data to a higher dimensional feature space using a Gaussian kernel, and performing regression with support vectors, quantification of alcohols was achieved with high accuracy (root mean squared error (RMSE) ~ 2-5%). Quantitative results of SORS-SVR method were used in decision trees where contaminated, sub-potent, or fair determinations were made with 90.7 % accuracy. In general, false negatives and false positives were samples that contained alcohols below the limit of detection, which varied for each alcohol type and the formulation (1-5 % v/v).”*

- b. Revise text from line 87 to 96. It is written to be more likely a part of the abstract or conclusion. The final part of the introduction should stress the proposed approach, the comparison performed with the golden standard routine and the overall advantage, not the results obtained! That will help the reader to understand the logical steps of the authors while is reading the results
- i. Proof of SORS advantage analyzing a mixture of contaminant with fixed %
 - ii. MA on the mixture on a different type of container (PE, PP glass)
 - iii. Comparison with a different parameter of the PLS, SVM models
 - iv. How is it achieve this insensitive to the container?? Is it the use of scale subtracted spectra? Need to be clearly stated

We have now revised the above mentioned part of the introduction as follows: *“Herein, a SORS-based quantitative method was developed for through-container analysis of hand sanitizer products. The superiority of SORS over traditional Raman was demonstrated using multivariate analysis (MVA) of spectra from hand sanitizer formulations with four different alcohols. When various types of containers (plastic, glass, opaque, transparent, etc.) were used for each formulation, only ~21% of the variation in the traditional Raman dataset was representative of contents (~77% of the variation was representative of containers). On the other hand, more than 99% of variation in the scaled-subtracted dataset was representative of contents, resulting in clear separation of alcohols in the score space of components representing the alcohols in the formulations. Previously, quantitative SORS has only been shown under container uniformity²¹, here a method was developed to quantify the amount of alcohol in hand sanitizer solutions inside containers with varying opacity and material type. Quantification was achieved using a normalization step prior to regression and taking advantage of concentration dependent effects in the Raman spectra of alcohol-water mixtures. By transforming the scaled-subtracted data to a higher dimensional feature space using a Gaussian kernel, and performing regression with support vectors, quantification of alcohols was achieved with high accuracy (root mean squared error (RMSE) ~ 2-5%). Quantitative results of SORS-SVR method were used in decision trees where contaminated, sub-potent, or fair determinations were made with 90.7 % accuracy. In general, false negatives and false positives were samples that contained alcohols below the limit of detection, which varied for each alcohol type and the formulation (1-5 % v/v).”*

- c. At the end of the introduction when the author presents their approach, they should refer to the limit of detection do Raman for the contaminant (concentration < 5 %) in comparison with the GS-MS. Is It enough for the analyzed system?

We thank the reviewer for allowing us to emphasize this point. In general, the goal in rapid screening is to identify priority samples for further testing. Although GC-MS or HPLC are benchmark methods in quantitative analysis, they are time consuming, tedious and are very rarely deployed in the field.

For hand sanitizers, identification of adulterated samples (such as methanol or 1-propanol substitution) and their expeditious removal from the supply chain is essential. While limits of detection (1-5% depending on the type of alcohol and matrix effects) in SORS is not comparable to single digit ppm levels that can be obtained in GC-MS, quantitative rapid screening enables identification of adulterated or out-of-specification (sub-potent or super-potent) samples.

In our revised manuscript, the final sentence in the introduction alludes to the goal of our work and the limit of detection: “*Quantitative results of SORS-SVR method were used in decision trees where contaminated, sub-potent, or fair determinations were made with 90.7 % accuracy. In general, false negatives and false positives were samples that contained alcohols below the limit of detection, which varied for each alcohol type and the formulation (1-5 % v/v).*”

d. Line 77 to 82, the author should add a reference supporting the claim of the problematic problem for the semi-quantitative model (due to self-absorption from both container or material itself)

In our revised manuscript, we added the following reference, where the limitation of SORS and transmission Raman spectroscopy with regard to self-absorption of surface and sub-surface molecules were discussed: “*Self-absorption corrected non-invasive transmission Raman spectroscopy (of biological tissue) by Gardner, Matousek and Stone from **Analyst**, 2021, **146**, 1260-1267*”.

e. Previous literature should be ref more properly, need to add references in the following statements:

i. Line 43 - Ref. previous work with the golden standard for purity testing of hand sanitisers (according to FDA, GS-MS) – Add a brief comparison with results that may be obtained with golden standard to highlight the advantage of the proposed approach.

We thank the reviewer for the comment. The proposed approach is not an alternative to the GC-MS methods for testing hand sanitizers. GC-MS methods are highly sensitive and can provide quantitative impurity analysis for a broad range of analytes. Currently, due to various matrix effects, GC-MS method development is still in progress and results are yet to be disseminated.

Spectroscopic methods deployed in the field with portable Raman spectrometers are complementary to the tedious and time-consuming GC/LC methods as they provide identification of highly adulterated samples. Our work takes a further step in quantification of certain alcohols and provides identification of out-of-specification samples. This was highlighted in our manuscript as “*Development of a rapid screening method may allow swift regulatory action or prioritization of samples for further analysis*”.

ii. Line 60-63 add a ref describing SORS principle and scale subtraction protocol. Refers properly the technique and the process first described by Matousek et al, therefore add the following:

- a. Mosca, S.; Conti, C.; Stone, N.; Matousek, P. Spatially offset Raman spectroscopy, *Nat. Rev. Methods Prim.* 2021, 1 (1), 21.
- b. Matousek, P. et al. Subsurface probing in diffusely scattering media using spatially offset Raman spectroscopy. *Appl. Spectrosc.* 59, 393–400 (2005).

We thank the reviewer for allowing us to cite previous work appropriately. In our revised manuscript, we added the references recommended by the reviewer.

2. The author should clarify within the text (results, method, SI..) what they mean with SORS data. Traditionally with the term, SORS means the Raman spectra acquired with spatial offset with respect to the excitation point. By reading the text I am not sure if the author instead with SORS means the scale subtraction between SORS and zero offsets. This must be clarified and eventually corrected.

We thank the reviewer for allowing us to clarify and correct the terminology used in our work. In our revised manuscript, we used “traditional Raman” for spectra collected at zero offset, “offset” for the spectra acquired with spatial offset, “scaled-subtracted” to represent the scaled subtraction between “offset” and “traditional Raman”. We have kept the use of the term SORS to only represent the technique “spatially offset Raman spectroscopy” and not “spatially offset Raman spectra”.

3. In the same way, it is not clear if the approach was applied separately to the zero offset and spatial offset spectra or the scaled subtracted spectra. Can the author retrieve the native spectra or just the scaled one?

We thank the reviewer for the comment. The native spectra was retrieved when needed, for instance, the “traditional Raman” and “scaled-subtracted” statistical comparison using MCR (Figure 3) was done by comparison of traditional (zero) spectra and scaled-subtracted spectra. Whereas the regression analysis was performed only on scaled-subtracted spectra.

4. Add if possible the information on the spatial offset (fixed, I suppose) used by the instrument. In the instrumentation section, it seems that a bunch of pre-processing were automatically applied to the acquired spectra (scaled subtraction and polynomial baseline subtraction), but then in the statistical analysis section seems that further pre-processing (baseline correction) is applied before Multivariate analysis. Please Clarify if MCR, PLS, SVM were applied on the scaled subtracted spectra or the raw acquired one. If so, take into account that the container contribution is always subtracted and how a fluctuation of thickness may affect the prediction of alcohol percentage.

We thank the reviewer for the comment. We have now added the fixed spatial offset used by the instrument. The reviewer is correct that when the scaled subtracted spectra is used, the container

contribution is subtracted and the change in the thickness of the container can affect the prediction result. For this very reason, additional preprocessing steps were taken, Whittaker filter was applied to remove artefacts from scaled subtraction (specifically in highly opaque containers) and spectra were normalized. Comments 11 and 12 by the reviewer #3 are also on the issue of container thickness, prediction methodology and preprocessing. We have provided a more detailed discussion in our responses to comments 11 and 12. In our revised manuscript, we have clarified that the scaled subtraction was used in statistical analysis and not the raw spectra.

5. In S1 - together with the container picture, add geometrical information of the cylinder (radius, height, thickness), useful information for understanding the eventual effect on photo migrations. In particular, add a rough range of the container thickness with a comment on how is the proposed model affected by a variation of container thickness. Since the author aimed for a universal method, this is an important aspect.

We thank the reviewer for this comment. In our revised supporting information, we have now included geometrical information of the containers and their thickness. In our analysis we found that thickness alone was not the sole reason affecting the accuracy of regression models. Material type and additives (glass, plastic, coloring additives, polymer molecular weight, cross-linking, etc.) were also critical factors. For instance, glass and gallon size plastic containers were among the thickest but were either transparent or semi-transparent in nature (indicating a lower molecular weight polymer and/or cross-linking) and samples in such containers could be accurately predicted. On the other hand, two samples that were prediction outliers were in highly dense plastic containers with average thickness, where the signal-to-noise ratio was significantly lower compared with other samples. In our revised manuscript, we have discussed this aspect: *Container material type (glass, plastic), thickness, additives used in the manufacturing (colorants, liners, etc.) and material processing parameters (polymer molecular weight, cross-linking, etc.) determine the transmittance of light and affect the Raman features and their intensity in the spectra during through-container spectroscopy. To minimize the intensity fluctuations and spectral distortions caused by variation in the containers, spectra were normalized during preprocessing (see Methods). Although, normalization alone cannot resolve the issue of quantification in through-container Raman spectroscopy, relative intensities of features could be utilized when more than one analyte is manifested in the spectrum.*”

6. Revise paragraph from line 152 to 155: the author used the wrong terminology for explaining the concept of SORS and sublayer sensitivity due to photon migration. The correct terminology should be used clearly. Specifically, It is not the incident light that is spread differently between the conventional Raman and SORS: if the excitation geometry is the same, the electromagnetic field and the intensity

distribution of incident light in the media is the same. What is different is the capability to collect photons that propagate deeper (but not the incident light!!!). With the zero spatial offsets, the majority of the collected photons are statistically the ones that propagate for shorter distance (therefore surface information). With the use of spatial offset, it is possible to detect a larger number of photons that travel statistically deeper in the media (subsurface + surface). I suggest the author be more clear and if they required a deep knowledge on the subject I suggest

reading the following reference on general photo migration (Martelli, F.; Binzoni, T.; Pifferi, A.; Spinelli, L.; Farina, A.; Torricelli, A. *Sci. Rep.* 2016, 6 (1), 27057.) or specifically SORS (Mosca, S.; Dey, P.; Tabish, T. A.; Palombo, F.; Stone, N.; Matousek, P. *Anal. Chem.* 2019, 91 (14), 8994–9000.)

We thank the reviewer for this critical correction. In our revised manuscript, we have corrected this terminology: *“In traditional Raman, light is collected at the source position where the majority of the photons are those that travel a short distance. Hence, sub-surface Raman features are usually overwhelmed by the surface fluorescence or Raman. For light collected at a lateral offset, photons that travel longer are likely to be detected as well. Therefore, inelastically scattered light collected at a spatial offset contains a higher portion of the sub-surface Raman photons compared with traditional Raman.”*

7. Specify in Figure 1.a, what the 2D- intensity distribution map refers to – is it a simulation of the collected photons..?

We thank the reviewer for this comment. We have now revised Figure 1A. The 2D map is intensity distribution of collected photons at positions indicated by green arrows. Red arrows indicate the incident light position.

8. The goal of the paragraph that starts from line 189 is not clear until the end of the paragraph (line 204 to 206). As it is, it is difficult to follow the presented results, I suggest to:

- a. Moving up (at the start of the paragraph) “spectra of binary mixtures of common contaminants and approved alcohols in hand sanitizer formulations in semi-transparent containers was shown”.
- b. to highlight in figure 2 the characteristic band of methanol, propanol and ethanol that were mentioned for the identification.

We thank the reviewer for their comment. In our revised manuscript, we have rearranged the paragraphs to aid the reader in following the presented results.

9. In SI, while introducing MCR, please mention the non-negativity constrain for the loadings – that’s why they are easily attributed to the components.

We thank the reviewer for their comment. We have mentioned non-negativity constraint in MCR in the revised version of our SI.

10. Figure 3, “MCR results”. The results not readable:

- a. panel from A-C) convert the 3D score plot into two 2D panels (eg. PC1 VS PC2, PC1 VS PC4 or a similar combination pc2 vs pc4 etc) will improve the vision of eventual clustering especially on the origin – place the legend on a side
- b. panel from D to E is related to the same MCR on the traditional Raman (it was not clear from the text and the sub-division) all the 7 components should be presented in the order of the same panel. The author should try to use a colour scale for the spectra that do not confuse the reader between the scores of the MCR and the actual measurements (dots in the scores plot), which are actually a combination of the different scores with the proper coefficients.
- c. Add the marks of the mentioned containers’ Raman peak in the loadings (ref to line260-261).
- d. In a similar way, panel C/F (related to MCR applied on SORS spectra) should contain all different loadings (7?) or just the 3 used for the score plot. I understand why the display also component 4, but the author should make a choice and keep the consistency. Eventually, the whole loadings and 2D scores plot can be added in SI.

We thank the reviewer for allowing us to improve data visualization. We have now added 2D scores plot in the SI. We have kept the 3D visualization for the manuscript as it allows decluttered representation of data (reduced number of figures).

We have combined the panels from the same MCR model in our revised manuscript and kept the color scale consistent for scores and components.

We added marks on component features that correspond to specific Raman bands of container materials and alcohols.

When the scaled-subtracted spectra is analyzed using MCR, a four component model was built and a cumulative fit of 99.16% was achieved. When the model was designed with seven components, components 5, 6, and 7 did not provide meaningful spectra and provided a combined fit less than 2%. Hence, a seven component MCR model was deemed ill-designed as it was not providing meaningful fit of the data.

11. The authors do not describe how the approach is quantitative. They clearly perform a calibration model with a specific % of Alcol but then, it is not clear what are markers used for the quantification. Do they use normalized intensity?

We thank reviewer 3 for allowing us to clarify our quantification methodology.

As reviewer 3 rightfully points out, all spectra were normalized by area. We calibrated our model with a carefully designed dataset, consisting of fixed alcohol concentrations with varying container materials and opacity, and varying alcohol concentrations in fixed opacity and material container. The design of this dataset allows the model to be trained for variations in both contents and containers. Our test dataset consisted of 173 samples, of which 53 were commercial products. The details of these datasets were listed in Table S1.

For quantification, we had developed several regression models in our work, all of these approaches were multivariate, where the entire spectra were utilized. Details of PLS regression algorithms were provided in the revised version of our manuscript and supporting information. For quantification, latent variables from PLS methods were demonstrated in Figure S8.

Figure S8. First 7 latent variables in the PLS model. Vertical dashed lines indicate unique alcohol features: 2-propanol at 818 cm^{-1} , 1-propanol at 859 cm^{-1} , ethanol at 882 cm^{-1} , and methanol at 1030 cm^{-1} .

12. At the end of the results/conclusion section, it was not discussed how the proposed approach manage to deal with a different container, the author simply stated the different accuracy in presence of more transparent or opaque media. Reading the abstract, I have expected a universal solution to the problem or a physical rationalize on the basis of the performed analysis to remove the container contributions. Instead, this is missing. The author should be careful about the claim stated in the abstract/introduction and conclusion. Moreover, the authors do not comment on relative intensity distortion due to absorption that can cause a fluctuation in the model, they do not refer to multiple problems that can't allow a direct quantitative analysis.

We thank reviewer 3 for allowing us to clarify how we have achieved container-universal quantitative detection.

As was promised in our abstract, our SVM coupled SORS method was able to identify and quantify the active ingredients (alcohols) of hand sanitizer formulations. Our method was universal in the sense that whether the containers were opaque or transparent, made of PET, PP, PE or glass, the alcohol content of hand sanitizer formulations inside could be predicted with high accuracy.

We thank the reviewer for reiterating the issue with container absorption, varying thickness, density, and other factors. Theoretically, a single analyte at varying concentrations in an environment where no matrix effect is present cannot be quantified accurately when container-induced intensity fluctuations are present. However, real-life products are more complex, as they generally include more than a single marker and the overall Raman spectra provide clues to matrix-related effects. Such is the case with hand sanitizers. To take advantage of such phenomena, in this work, spectra were first normalized. Normalization does not resolve irregularities in the spectra due to container-related intensity variations, however, it generally allows multi-alcohol mixtures to be predicted reasonably well, using relative intensities. This was clearly demonstrated in our revised manuscript, using a PLS model with loadings representing separation between all four alcohols. Yet, this model (PLS-4) yielded larger prediction errors for single-alcohol formulations due to the lack of a relative Raman feature (Figure 4A).

Figure 4A (Right Panel). Predictions with four latent variable PLS were less accurate for 1-alcohol formulations.

When additional latent variables were analyzed (Figure S8), variations caused by spectral shifts in alcohol-water mixtures were encountered (Spectral shifts and their concentration dependence was documented in Figure 2 and Figure S2).

Figure S2 (Top). Measured spectral shifts of C-O bands of alcohols used for hand sanitizer formulations in this study when the water-alcohol amounts were varied.

Figure 4B (Right Panel). Predictions with eight latent variable PLS improved prediction accuracy by taking advantage of matrix effects in single-alcohol formulations. 1-propanol based samples (where peak shifts were less obvious) remained challenging.

Therefore, including additional latent variables improved the PLS model however for alcohols where spectral shifts were less prominent, predictions were still poor and samples in transparent and semi-transparent containers were overpredicted while samples in opaque containers were slightly underpredicted.

Support vector machines (SVM) can be used to solve regression problems. Using the kernel trick, SVM regression can be effective for problems where nonlinear phenomena occur. We have demonstrated that while matrix effects could be utilized for container-universality, they were unique for each alcohol and its dependence on the quantity of alcohol in the formulation was nonlinear. Hence, SVR with a Gaussian radial function kernel, was an excellent candidate for the container-universal through-container testing of hand sanitizers. SVM based regression was explained in detail in the supporting information. SVM is a machine learning algorithm that differs from PLS and PCR regression methods in various ways.

Figure 4C (Middle Panel). Predictions from the SVM regression. Various outliers (red squares) and well-predicted spectra were compared below.

Briefly, the regression problem in SVM is no longer trivial ($y = (\omega^t x_i) + b$) and the vectors that yield predictions are not necessarily representative of spectral features like the loadings in PCA or latent variables in PLS.³ In SVM, a parameter called cost is optimized for minimal cross-validation error and predictions are generated. An additional parameter for the basis function can also be optimized using a grid search (Figure S10). In our revised manuscript, we have identified several outliers to our SVM based prediction model (red squares in Figure 4C-middle panel). We have also identified products that were predicted poorly using PLS-4 but well-predicted using SVM (black squares in Figure 4C-middle panel). Spectra from the abovementioned points were plotted before and after preprocessing (normalization) (Figure S6) and matrix effects, signal-to-noise ratio and other container-related intensity fluctuations were discussed.

Figure S6. Scaled-subtracted spectra of well (A) and poorly (B) predicted hand sanitizer products, before (top) and after (middle) normalization. Spectra in top and middle panels are offset for clarity. Spectral shifts (bottom) are resolved in (A) where prediction errors are minimal, whereas in (B) spectra is noisy and does not allow resolution of shifts.

In our revised manuscript, we added the following discussion on this issue: “*To confirm, spectra from well (smaller prediction error) or poorly (larger prediction error) predicted hand sanitizer products (highlighted in black or red squares in Figure 4, respectively) in transparent, semi-transparent, and opaque containers were selected. Spectra were compared before and after preprocessing (normalization and baseline correction with Whittaker filter, Figure S10). For well-predicted products, the spectral shifts*

due to hydrogen bonding in alcohol-water mixtures were clearly resolved. For products with larger prediction errors, the signal-to-noise ratio (SNR) was not large enough for the C-O band shift to be resolved clearly. It should be noted that reduced SNR in certain products could not be attributed to a single property of the containers as no direct relationship could be deduced based on thickness or material type.

There were also two products in transparent containers that were poorly predicted with SVR, which contained a total alcohol concentration of ~70%, consisting of about equal amounts of ethanol and 2-propanol. This error was expected, since the models were not trained for ethanol and 2-propanol mixtures, as it is unusual to use both of the recommended alcohols in a single hand sanitizer formulation. The error for these products were less in PLS models, where loadings of several LVs contained separation for ethanol and 2-propanol. Note that the a separate SVR model was computed for each alcohol used in the study, which is a limitation of the SVM method that can be addressed with training dataset design.”

1. USP, <858> RAMAN SPECTROSCOPY. Official Date 1-Nov-2020.
2. Olds, W. J.; Sundarajoo, S.; Selby, M.; Cletus, B.; Fredericks, P. M.; Izake, E. L., Noninvasive, Quantitative Analysis of Drug Mixtures in Containers Using Spatially Offset Raman Spectroscopy (SORS) and Multivariate Statistical Analysis. *Appl. Spectrosc.* **2012**, *66* (5), 530-537.
3. Drucker, H.; Burges, C. J.; Kaufman, L.; Smola, A.; Vapnik, V., Support vector regression machines. *Advances in neural information processing systems* **1996**, *9*, 155-161.

REVIEWERS' COMMENTS:

Reviewer #2 (Remarks to the Author):

I have reviewed the author's revision to the manuscript and am satisfied with their corrections. I recommend that this manuscript be published in its current form.

Reviewer #3 (Remarks to the Author):

The authors have addressed most of the raised points. I think they have highly improved the clarity and the readability of their work. I think the paper is highly technical and well written. However, my main concern on the overall moderate novelty remains. I think this work does not match the novelty criteria for publication in Communication Chemistry but it will suit better a more specialized journal.

Response to Reviewers

Reviewer #2 (Remarks to the Author):

I have reviewed the author's revision to the manuscript and am satisfied with their corrections. I recommend that this manuscript be published in its current form.

We thank the reviewer for their feedback and contributions.

Reviewer #3 (Remarks to the Author):

The authors have addressed most of the raised points. I think they have highly improved the clarity and the readability of their work. I think the paper is highly technical and well written. However, my main concern on the overall moderate novelty remains. I think this work does not match the novelty criteria for publication in *Communication Chemistry* but it will suit better a more specialized journal.

We thank reviewer #3 for technically elevating our work and improving its clarity.

The novelty of our work is now addressed in our manuscript with the following revisions:

“Overall, a chemometric approach was utilized in SORS to enable quantification via through-container screening, regardless of container thickness, material, or opacity. Overcoming the limitations (lack of specificity and consideration for matrix effects in finished products) of widely used library search methods, SORS-SVR provided non-destructive and quantitative analysis of medical countermeasure products used against COVID-19.”

“Using spatially offset Raman spectroscopy (SORS) and support vector regression (SVR), active ingredients (e.g., type of alcohol) of 173 commercial and in-house products were identified and quantified regardless of the container material or opacity.”

Briefly, through-container quantification regardless of container type, material and opacity has not been demonstrated before. Our work is not only a demonstration but also an application of our method to real-world products. In addition, SORS-SVR can be used in various other medical countermeasures as well as hand sanitizers where quantitative and non-destructive characterization is critical.